# Zero-shot Meta-learning for Tabular Prediction Tasks with Adversarially Pre-trained Transformer

Yulun Wu [1]   Doron L. Bergman [2]

## Abstract

We present an Adversarially Pre-trained Transformer (APT) that is able to perform zero-shot meta-learning on tabular prediction tasks without pre-training on any real-world dataset, extending on the recent development of Prior-Data Fitted Networks (PFNs) and TabPFN. Specifically, APT is pre-trained with adversarial synthetic data agents, who continue to shift their underlying data generating distribution and deliberately challenge the model with different synthetic datasets. In addition, we propose a mixture block architecture that is able to handle classification tasks with arbitrary number of classes, addressing the class size limitation – a crucial weakness of prior deep tabular zero-shot learners. In experiments, we show that our framework matches state-of-the-art performance on small classification tasks without filtering on dataset characteristics such as number of classes and number of missing values, while maintaining an average runtime under one second. On common benchmark dataset suites in both classification and regression, we show that adversarial pre-training was able to enhance TabPFN's performance. In our analysis, we demonstrate that the adversarial synthetic data agents were able to generate a more diverse collection of data compared to the ordinary random generator in TabPFN. In addition, we demonstrate that our mixture block neural design has improved generalizability and greatly accelerated pre-training.

## 1. Introduction

In standard deep learning workflows, models are either trained per dataset, or employed on data in a form compatible with, and drawn from, the same distribution as the datasets it was previously trained on. Even in transfer learning, where the target of the model is changed, the input is at most expanded, but at least overlaps heavily with the data distribution that the model has previously seen in training. This is in contrast with meta learning (Finn et al., 2017; Nichol & Schulman, 2018; Lemke et al., 2015; Vanschoren, 2018; Feurer et al., 2022; Hospedales et al., 2021; Zintgraf et al., 2021), where a model is trained to be adaptive to new datasets such that few gradient updates or fine-tuning are needed, instead of training a new model specialized to every distinct dataset from scratch. In meta learning, rather than modeling a specific dataset, the model is trained to learn how to learn. This has multiple advantages. First, meta learning is highly adaptable (Huisman et al., 2021; Finn et al., 2017; Frans & Witkowski, 2021) – it learns more generalized representations that can be adapted to new tasks and different domains. Second, meta learning makes efficient use of data (Finn et al., 2017; Gevaert, 2021) – it supports learning from just a few samples. Third, as a consequence of its efficient use of (small) data, the model can reach a point where it is able to make meaningful predictions very quickly (Vanschoren, 2018).

In prior work, Verma et al. (2020) discussed the notion of zero-shot meta-learning. They train a generative adversarial network conditioned on class attributes, that can generate novel (previously unseen) class samples. This relies on the inputs present in the training data (class attributes) to be indicative of the new unseen classes. While they do not use gradient updates on the unseen data for prediction, they rely on the input data coming at the very least from a very similar distribution to that of the training data. The scope of problems this work aims to address is pristine zero-shot meta learning: given an unseen dataset from an unseen task after the model is pre-trained and deployed, can we do prediction on this dataset without training the model on it? Specifically, with zero gradient update on the model, and with no reliance on the context similarity between this dataset and the datasets that the model was pre-trained on. Note that this concept of zero-shot is slightly different from that in large vision and language models (Mann et al., 2020; Perez et al., 2021; Tsimpoukelli et al., 2021; Cahyawijaya et al., 2024; Ahmed & Devanbu, 2022) – the unseen datasets can

---

[1]University of California, Berkeley [2]Capital One. Correspondence to: Yulun Wu <yulun_wu@berkeley.edu>.

*Proceedings of the 42nd International Conference on Machine Learning*, Vancouver, Canada. PMLR 267, 2025. Copyright 2025 by the author(s).

entail heterogeneous fields or class labels that were never observed during pre-training, and zero-shot in this context refers to the amount of model optimization conducted being zero given the unseen dataset rather than the amount of empirical examples seen being zero. The advantage of successfully establishing such a model is the exceptional generalizability and runtime.

A few recent breakthroughs (Müller et al., 2021; Hollmann et al., 2022) have demonstrated that achieving this aspiration is possible: Müller et al. (2021) introduced Prior-Data Fitted Networks (PFNs). They pursue zero-shot meta-learning by using transformers pre-trained on synthetic data generated from a collection of prior distributions, to perform approximate Bayesian inference using in-context learning (Luo et al., 2018; Mann et al., 2020). PFNs do not fit a model on downstream training data, instead feeding training data into the context in forward pass and making predictions conditioned on the context. Hollmann et al. (2022) introduced a PFN specifically aimed at tabular datasets – TabPFN. A more detailed background review on PFNs and specifically TabPFN can be found in Appendix A. Tabular data – data organized in rows and columns, and characterized by an unlimited heterogeneity of data fields, remains an area of machine learning where deep neural networks (DNNs) still struggle (Borisov et al., 2022; Shwartz-Ziv & Armon, 2022; McElfresh et al., 2024; Ye et al., 2024b) to push the boundaries of the state-of-the-art gradient boosted decision trees (GBDTs) (Prokhorenkova et al., 2018; Chen & Guestrin, 2016; Ke et al., 2017), despite numerous approaches (Borisov et al., 2022; Somepalli et al., 2021; Grinsztajn et al., 2022; Gorishniy et al., 2021; Rubachev et al., 2022; Levin et al., 2022; Kadra et al., 2021a; Arik & Pfister, 2021; Popov et al., 2019). Yet, tabular data is one of the most common data types in real-world machine learning (ML) applications (Chui et al., 2018; Borisov et al., 2022; Shwartz-Ziv & Armon, 2022). Although TabPFN has demonstrated exceptional zero-shot meta-learning capability on certain small tabular prediction tasks, we show that the distribution of synthetic data used in its pre-training is actually quite limited. Besides, the class size constraints of TabPFN pose a significant limitation on its generalizability – this might not be an important concern for the traditional one-model-for-one-domain pipeline, but is a crucial weakness for a zero-shot meta-learner (ZSML) since an unprecedented number of class labels could be present in inference time. Note that zero-shot meta-learning is largely similar to foundation modeling but slightly different in its scale and objective – it does not necessarily involve billions of parameters to learn the distribution of data and acquire token representations in a broad domain such as language or health records, but to model the general prediction logic and learn how to acquire data representations in unseen domains during inference time.

Similar to Hollmann et al. (2022), we investigate the capa-

bility of zero-shot meta-learning under the scope of tabular data prediction problems. Our contributions are listed as follows:

- We propose an adversarial synthetic data pre-training approach on PFNs to establish a zero-shot meta-learner that is able to handle tabular prediction tasks with improved performance.

- We eliminated the class size limitation for TabPFN on classification tasks by proposing the mixture block neural design, which yields a zero-shot meta-learner with better generalizability.

- In experiments, we show that our framework achieves state-of-the-art performance on small tabular classification tasks without filtering on class size, feature size, number of categorical features or number of missing values, and improved upon TabPFN in both classification and regression. We show that the adversarial data agents are able to enrich the synthetic data generating distribution, and the mixture block is able to generalize to unseen class size and accelerate pre-training.

## 2. Proposed Method

Our Adversarially Pre-trained Transformer (APT) model is pre-trained once offline using a mix of random synthetic data generators and adversarial synthetic data agents. In this phase, the goal of the model is not to learn the specific pattern or probability distribution of any given dataset, but to learn the general prediction logic and means to represent various data, i.e. learning to learn. Once pre-trained and deployed, the model makes predictions on the testing set of any real-world dataset of interest in one forward pass, without performing any back-propagation or gradient updates of its weights. A demonstration of the workflow is shown in Figure 1. In Section 2.1, we describe the adversarial data agents in detail, whose goal is to continuously produce diverse and more challenging datasets for the meta-learning model during pre-training; in Section 2.2, we elaborate on the architecture of our transformer model, which has no restrictions on the class size of any real-world datasets practitioners provide.

### 2.1. Adversarial Data Agents

In the pre-training phase, we compose a batch of $m$ datasets $\{X^{(k)}, \boldsymbol{y}^{(k)}\}_{1 \leq k \leq m}$ in each iteration using $m$ different data generators $\{g_1, \ldots, g_m\}$ that each independently generate $n$ number of data points, where $X^{(k)} = [\boldsymbol{x}_i^{(k)}]_{i \leq n}^{\top} = [x_{i,j}^{(k)}]_{i \leq n, j \leq d_k}$ and $\boldsymbol{y}^{(k)} = [y_i^{(k)}]_{i \leq n}^{\top}$ are the predictor matrix and response vector (denoted as $X$ and $\boldsymbol{y}$ when no index is specified) with feature size $d_k$. We adopted the multi-layer perceptron (MLP) construction introduced in

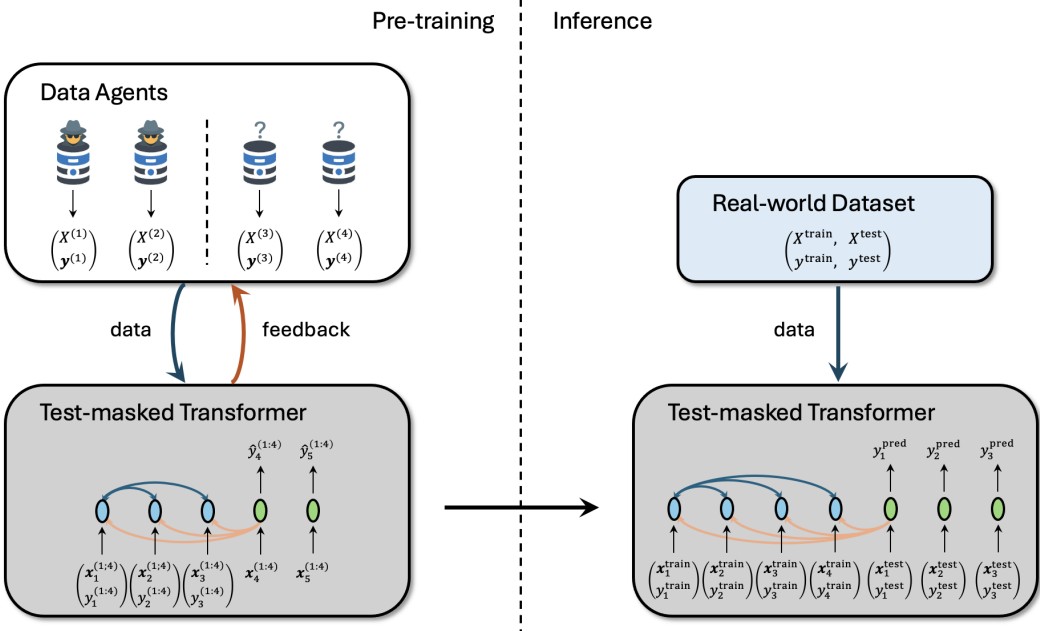

Figure 1. The model workflow of Adversarially Pre-trained Transformer (APT). Pre-training is done once, offline, with datasets generated by a mix of random synthetic data generators and adversarial synthetic data agents. The train-test split is randomly sampled for each batch of datasets. After the model is pre-trained and deployed, predictions are done per real-world dataset, online, with one forward pass and no parameter update. The transformer is test-masked, meaning that each token only attends to training data tokens. For cleanliness of the figure, only the attentions to and from the first training data token and the first testing data token are plotted.

Hollmann et al. (2022) for each generator instance, where predictors $\boldsymbol{x}_i^{(k)}$ and response $y_i^{(k)}$ are values of randomly selected neurons in sparsified noisy MLPs with some additional pre-processing. More details regarding this approach can be found in Appendix A.1.

Different from Hollmann et al. (2022), instead of generating datasets solely from randomly initialized sparse MLPs, a subset of the $m$ generators in our framework are adversarial agents that learn from the model's performance on the generated data, and perform gradient *ascent* on the model's prediction loss. In other words, these adversarial agents challenge the model by constantly shifting the synthetic data generating distributions to deliberately produce datasets that are more difficult for the model to handle. The loss for an adversarial agent $g_\eta$ with respect to prediction model $q_\theta$ can be written as

$$\mathcal{L}(g_\eta) = \mathbb{E}_{X, \boldsymbol{y} \sim g_\eta} \log q_\theta(\boldsymbol{y}_{(l+1):n} | X_{(l+1):n}, \{X_{1:l}, \boldsymbol{y}_{1:l}\}) \tag{1}$$

where $\{X_{1:l}, \boldsymbol{y}_{1:l}\}$ and $\{X_{(l+1):n}, \boldsymbol{y}_{(l+1):n}\}$ are the training and testing set split from generated dataset $\{X, \boldsymbol{y}\}$ at position $l$. In the following sections, we refer to the former (generators based on randomly initialized MLPs) as ordinary data generator, and the latter (generators based on adversarially updated MLPs) as adversarial data agents.

**Relation to Classic Adversarial Training**  In relation to GANs (Goodfellow et al., 2014), the data agents here are the generators, and the meta-learner is the discriminator. Contrary to classic adversarial training, there is no real versus fake samples for the discriminator to distinguish in this context. The generator (data agent) and the discriminator (meta-learner) have one coherent competing objective: the meta-learner seeks to minimize the prediction loss on data generated by the data agents, while the data agent seeks to generate data that maximize the prediction loss by the meta-learner. As a result, the desired gradients for updating the discriminator is but a flip of sign to its gradients calculated through back propagation on the generator's objective. Hence, both the meta-learner and the data agents can be updated in one single iteration after loss calculation in this scenario. This results in a more efficient adversarial training, and we further reduce its potential of mode collapse with data agent reset described in the last paragraph of this section. Note that contrary to classic GANs, the discriminator is the final product in this context rather than the generator.

**Discretization of Variables**  A key challenge in establishing adversarial data agents is the gradient flow under discretization: how do we generate synthetic data with categorical features while being able to perform end-to-end loss back-propagation? Inspired by the Gumbel-Softmax

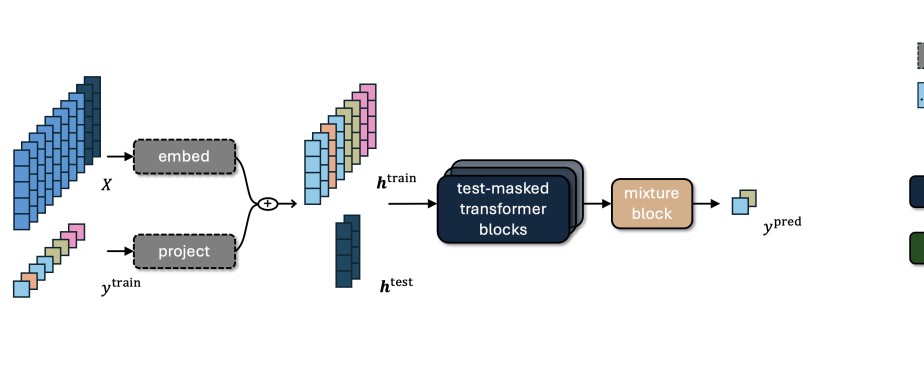 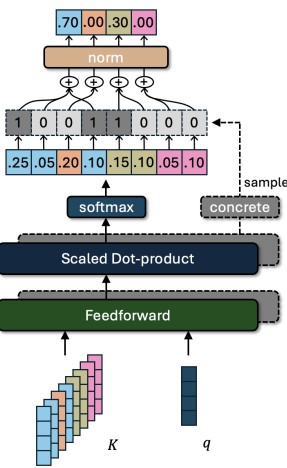

(a) Model architecture for classification tasks      (b) Mixture block

*Figure 2.* Model architecture and the mixture block. a) $X = (X^{\text{train}}, X^{\text{test}})$ and $\boldsymbol{y}^{\text{train}}$ are embedded on $\mathbb{R}^{d_{\text{model}}}$ using a feature embedding block and linear projection respectively. Then, embeddings for $X^{\text{train}}$ and $\boldsymbol{y}^{\text{train}}$ are added as $\boldsymbol{h}^{\text{train}}$, embeddings for $X^{\text{test}}$ are denoted as $\boldsymbol{h}^{\text{test}}$. Embeddings ($\boldsymbol{h}^{\text{train}}, \boldsymbol{h}^{\text{test}}$) are then passed to the transformer blocks with attention towards test embedding $\boldsymbol{h}^{\text{test}}$ masked, same as Hollmann et al. (2022). Finally, the outputs from transformer blocks are transformed to class probabilities through the mixture block for classification tasks, or directly transformed to point predictions through standard dense final layer for regression tasks. b) For each data point in the testing set, we use its output $q$ after transformer blocks to query training data's outputs $K$. With two different dense feedforwards, two sets of logits are predicted: one set of logits are used to calculate the scaled softmax probabilities – these probabilities indicate how likely that the testing point is in the same class as the corresponding training points; the other set of logits are used to sample soft-discrete binary gates via Concrete distribution to sparsify these probabilities. Finally, the gated probabilities from the same class are added together to yield the final predictions.

trick (Jang et al., 2016) and the Concrete distribution (Maddison et al., 2016), we propose a continuous relaxation of discretization that naturally extends on the ranking discretization approach introduced in Hollmann et al. (2022), controlled by a user-specified temperature hyperparameter $\tau$. For the $j$-th feature column $\boldsymbol{x}_{\cdot,j}$ of a predictor matrix $X$ and the corresponding $N_j - 1$ randomly sampled Gaussian quantiles $Q_j^{(1)} < Q_j^{(2)} < \cdots < Q_j^{(N_j-1)}$ at the initialization of the corresponding data agent, the soft-discretization that converts the $i$-th value of the $j$-th feature $x_{i,j}$ to a soft-categorical value with cardinality $N_j$ is given by

$$
x_{i,j}^{cat} = \pi\left(\left|\left\{x_{i,j} \geq \tilde{Q}_j^{(l)}\right\}_l\right|\right) +
$$
$$
\tau \cdot \log\left(1 + \frac{x_{i,j} - \tilde{Q}_j^{\left(\left|\left\{x_{i,j} \geq \tilde{Q}_j^{(l)}\right\}_l\right|\right)}}{\tilde{Q}_j^{\left(1+\left|\left\{x_{i,j} \geq \tilde{Q}_j^{(l)}\right\}_l\right|\right)} - \tilde{Q}_j^{\left(\left|\left\{x_{i,j} \geq \tilde{Q}_j^{(l)}\right\}_l\right|\right)}}\right)
$$
$$(2)$$

where $\pi$ is a permutation function on integer domain $\{1, 2, \ldots, N_j - 1\}$, $\tilde{Q}_j^{(l)} = \mu(\boldsymbol{x}_{\cdot,j}) + \sigma(\boldsymbol{x}_{\cdot,j}) \cdot Q_j^{(l)}$ for $1 \leq l < N_j$ are the unnormalized quantiles with boundaries $\tilde{Q}_j^{(0)} = \min(\boldsymbol{x}_{\cdot,j})$ and $\tilde{Q}_j^{(N_k)} = \max(\boldsymbol{x}_{\cdot,j})$, and $|\{v \geq \tilde{Q}_j^{(l)}\}_l| = \sum_l I(v \geq \tilde{Q}_j^{(l)})$ is the position of a value $v$ in the ordered sequence $\{\tilde{Q}_j^{(l)}\}_{1 \leq l \leq N_j}$. A visual demon-

stration of this conversion can be found on the right side of Figure 6 in the Appendix. Same as Hollmann et al. (2022), the extended ranking discretization approach decides the value of a categorical variable using only the continuous scalar $x_{i,j}$, i.e. the value of one neuron in the sparsified noisy MLP, as opposed to the Gumbel-Softmax or Concrete distribution approach which would require selecting $N_j$ neurons as logits of the $N_j$ classes. In our early experiments, we found that sampling multiple neurons to decide the value of one categorical feature achieved significantly worse performance than ranking discretization. Furthermore, since we do not desire to learn the explicit form of these distributions, explicitly generating class logits is not a necessity, and hence we prefer a more efficient differentiable discretization technique that does not involve reparameterization tricks, softmax operations or excessive samplings.

**Data Agent Reset** In terms of the diversity of generated data, there is a balance between adversarially updating the neurons in the MLPs and re-initializing the MLPs all together. Although in the short run, re-initializing the MLPs and the corresponding random factors (number of features, number of classes, etc.) instantaneously yield new datasets with a high chance of possessing much different fields and distributions from the previous, such generation is constrained by the domain of distribution defined by the preset

range of hyperparameters in the long run (we show some evidence on this in Section 3.2). On the other hand, although adversarial data agents are performance-driven and could explore out-of-distribution regions better than random initialization, it also has the potential to converge to the Nash equilibrium and reach a stalemate with the meta-learner – for example, converging to a state where generated predictors $x$ and response $y$ have no correlation. Hence, we combine the two approaches and reset the adversarial data agents every $N_e$ epochs to avoid such convergence. To speak from the GANs angle, we are letting the discriminator, i.e. the meta-learner, to periodically gain an advantage and slightly beat the generator. Different from classic GANs, the discriminator is the desired model here while the generator is the supporting entity, hence exploration is more important than optimization for the generator in this context.

## 2.2. Mixture Block Architecture

Contrary to modern DNNs, traditional ML algorithms such as K-nearest neighbors and tree-based methods are more flexible in terms of their ability to handle varying cardinality of classification labels, in the sense that they do not entail fixed-size dense layer parameters that cannot generalize to a different classification task with different label cardinality. This is not much of an issue for the traditional one-model-for-one-dataset ML pipeline, but is of significant importance for zero-shot meta-learners, yet unaddressed in prior works. Inspired by how tree-based methods solve classification tasks in a manner that is compliant to the empirical values and cardinality of training labels, we propose a scatter-sum mixture block as the output prediction head for classification tasks that significantly departs from the ordinary dense final layer approach. A visual demonstration can be found on the right of Figure 2. For each data point in the testing set, we use its embedding after the transformer blocks to query the embeddings of training data, and yield two sets of logits via two separate feedforwards: one set of logits is used to calculate softmax probability weights of keys and the other set is used to sample soft-discrete gates via Concrete distribution (Maddison et al., 2016) to sparsify these weights. In essence, these gates govern the splits of training data in relation to the testing query, such that the final prediction only pays attention to a subset of relevant training data representations. In our preliminary experiments, we discovered that sparsifying attention through these gates are crucial to performance, and the mixture block works poorly without this component. The output class probabilities are then acquired by a scatter summation of non-gated values using their original labels as index. Relating to tree-based methods, the gates here are used to determine the subset of training data that are in the same split of leaf nodes as a given testing data point, and the weights are used to determine the relative importance of each label in that split.

Contrary to tree-based methods, the splits are point-specific, i.e. there is a different split decided for each testing data point, and the decision within the split is weighted rather than via majority voting. Note that this approach does not change the order of computation complexity in terms of data size and data dimensions – it simply removes the final dense layer and adds two more multi-head attentions and feedforwards to the transformer architecture in a non-sequential manner.

**Large Data Size and Feature Size** Compared to the class size limitation, the feature size limitation of PFNs is relatively less tricky in theory, and there are already a few straightforward solutions concurrent with this work (Hollmann et al., 2025; Qu et al., 2025) that extend TabPFN's capabilities in handling datasets with larger number of features, as well as larger number of samples. Besides, the data capacity of PFNs could be adequately expanded by incorporating some of the recent advancements in general transformer and state-space model research (Wu et al., 2022; Bulatov et al., 2023). Therefore, we do not put emphasis on addressing these problems in this work, and make two simple adaptations to APT based on patch embedding and batch aggregation in the event that prediction on large datasets is required. See Appendix B for details. Note that concurrent solutions such as Hollmann et al. (2025); Qu et al. (2025) do not pose conflict with our proposed architecture (mixture block only modifies the last layer of the model), thus can be naturally incorporated into our framework as the practitioners desire.

## 3. Experiment

We evaluated our model and competing algorithms on common ML benchmarking dataset suites for tabular classification and tabular regression problems. In Section 3.1, we show that APT achieves state-of-the-art performance on small tabular classification tasks with a runtime comparable to that of TabPFN. In Section 3.2, we present qualitative analysis on the impact and characteristics of the adversarial data agents. In Section 3.3, we demonstrate the generalizability of the mixture block and its effect on pre-training. In Section 3.4, we provide ablation study and show that adversarial pre-training was able to enhance the performance of TabPFN on both classification and regression tasks.

**Datasets** For classification, we used the curated open-source OpenML-CC18 dataset suite (Bischl et al., 2021) containing 68 popular tabular benchmark datasets (4 vision datasets *mnist_784*, *CIFAR_10*, *Devnagari-Script*, and *Fashion-MNIST* are not treated as tabular and removed from the total 72 datasets), and our main results are presented on all small datasets (number of samples no larger than 2,000) in OpenML-CC18 similar to Hollmann et al. (2022),

*Table 1.* Performance of algorithms on 35 small datasets with no larger than 2,000 data points in the OpenML-CC18 suite, given one hour of time budget. Note that there are two styles of standard deviation (std.) calculation for AUC: 1) first take the mean of AUC across datasets, then calculate the std. across splits (std. of mean), as used by TabPFN (Hollmann et al., 2022); 2) first calculate the std. across splits on each dataset, then take the mean across datasets (mean of std.), as used by TabZilla (McElfresh et al., 2024). Our result table largely adopted the style of TabZilla, but we present both std.'s here for clarity. The std. of mean shows variation on suite level, which is more likely to result in a statistical significance compared to mean of std., which shows average variation on dataset level. The mean of AUC taken across splits are used as the scoring metric to calculate "Rank" and "Wins" of each algorithm across datasets. If many algorithms are tied for first, a win is assigned to each first-place algorithm. Same as TabZilla (McElfresh et al., 2024), the table is ordered by the mean of rank. The full results on each dataset for top algorithms are shown in Table 5 of Appendix D.

| | Rank ↓ | | | | ROC-AUC ↑ | | | Wins ↑ | Time (sec.) ↓ (Tune + Train + Predict) | |
|---|---|---|---|---|---|---|---|---|---|---|
| | mean | med. | min | max | mean | std. of mean | mean of std. | num. | mean | med. |
| APT | **3.86** | **3** | **1** | 11 | **0.921** | 0.003 | 0.019 | **13** | 0.90 | 0.40 |
| CatBoost | 4.03 | 4 | **1** | **9** | 0.918 | 0.002 | 0.020 | 6 | 3542.42 | 3555.74 |
| TabPFN | 4.57 | 4 | **1** | 11 | 0.913 | 0.003 | 0.020 | 4 | **0.86** | **0.37** |
| SVM | 4.89 | 4 | **1** | 12 | 0.904 | 0.003 | 0.023 | 10 | 1175.58 | 481.50 |
| XGBoost | 5.37 | 5 | **1** | 10 | 0.914 | 0.006 | 0.020 | 4 | 3607.78 | 3598.91 |
| LightGBM | 5.60 | 6 | **1** | 11 | 0.917 | 0.003 | 0.019 | 3 | 3542.94 | 3582.07 |
| LASSO-Logistic | 6.69 | 8 | **1** | 12 | 0.908 | 0.001 | 0.023 | 3 | 1519.41 | 1227.52 |
| Ridge-Logistic | 6.91 | 8 | **1** | 11 | 0.907 | 0.001 | 0.022 | 1 | 1479.93 | 845.59 |
| RandomForest | 7.17 | 7 | **1** | 12 | 0.908 | 0.003 | 0.021 | 3 | 1736.71 | 1476.37 |
| ResNet | 7.69 | 9 | **1** | 12 | 0.825 | 0.004 | 0.040 | 3 | 3582.15 | 3597.41 |
| KNN | 9.57 | 11 | **1** | 12 | 0.884 | 0.006 | 0.024 | 1 | 127.82 | 77.31 |
| SAINT | 9.97 | 12 | **1** | 12 | 0.759 | 0.017 | 0.077 | 1 | 3597.41 | 3594.41 |

except that 1) there is no additional filtering, i.e. all datasets regardless of number of classes, number of features, number of categorical features, and number of missing values are kept in our evaluation pool, composing a more general collection of datasets. This brings the number of datasets in the evaluation pool from 18 to 35; 2) The train-test split is set to 80-20 instead of the unconventional 50-50. For regression benchmarking, we used the curated open-source OpenML-CTR23 dataset suite (Fischer et al., 2023).

**Algorithms** We compared APT to the top 3 GBDT algorithms (CatBoost (Prokhorenkova et al., 2018), XGBoost (Chen & Guestrin, 2016), LightGBM (Ke et al., 2017)) and the top 3 DNN methods (TabPFN (Hollmann et al., 2022), Tabular ResNet (Gorishniy et al., 2021), SAINT (Somepalli et al., 2021)) in the main experiments of TabZilla (McElfresh et al., 2024), as well as 5 standard machine learning algorithms (KNN (Cover & Hart, 1967), Ridge (Tikhonov, 1963), LASSO (Tibshirani, 1996), SVM (Cortes, 1995), Random Forest (Ho, 1995)).

**Hyperparameters** The hyperparameter search space of benchmark models is directly inherited from Hollmann et al. (2022), and directly inherited from McElfresh et al. (2024) if the benchmark model is not in Hollmann et al. (2022). TabPFN is pre-trained with hyperparameters directly inherited from their released checkpoint, only changing the maximum number of classes from 10 to 26, which is the

maximal class size of datasets in the OpenML-CC18 suite. For APT, all common hyperparameters shared with TabPFN are directly inherited from TabPFN. See Appendix C for more details. A total of 12.5% of the data generators are adversarial data agents during the pre-training of APT, with learning rate $10^{-1}$, weight decay $10^{-5}$, soft-discretization temperature $10^{-2}$, and 2,000 gradient steps between resets.

### 3.1. APT Achieves State-of-the-art Performance on Small Tabular Classification Tasks

We evaluated APT and benchmark models on small datasets in OpenML-CC18 using area under the receiver operating characteristic curve (ROC-AUC) with the one-vs-one (OVO) multi-class evaluation configuration, similar to Hollmann et al. (2022). Previously, Hollmann et al. (2022) has shown that TabPFN matches the performance of state-of-the-art GBDT algorithms and outperforms them on small datasets that have less than 100 features, less than 10 classes, no categorical features, and no missing values in their main results. In this work, we do not impose any of these restrictions to further examine APT's and TabPFN's zero-shot meta-learning capability. The results are presented in Table 1. For datasets with number of features larger than 100, we subsample 100 features similar to (McElfresh et al., 2024). In these experiments, APT achieved state-of-the-art performances with a runtime similar to that of TabPFN. The average runtime of APT increased by 4.6% compared to

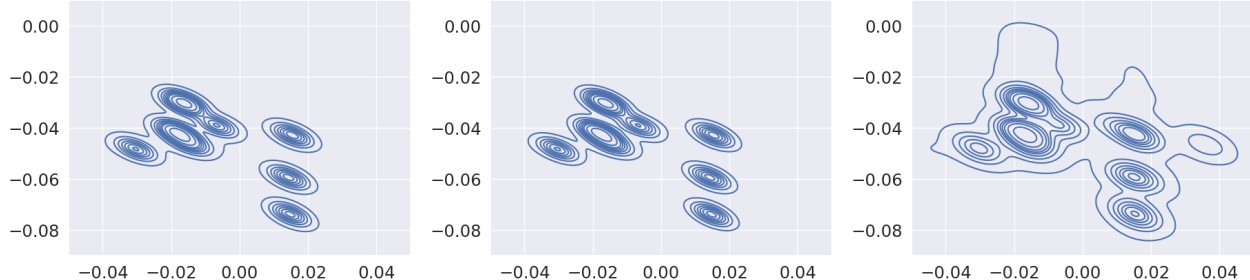

(a) Distribution of data in datasets generated by a set of ordinary data generators

(b) Distribution of data generated by another independent set of ordinary generators

(c) Distribution of data generated by a set of adversarial data agents

*Figure 3.* Contour plot of two-dimensional data generated by ordinary data generators and adversarial data agents. Each subplot contains a total of 100,000 data points from 2,000 datasets. Note that subplot (a) and subplot (b) are two independent sets of ordinary generators with no mutual, as each dataset is generated by an independently initialized random sparse neural network. Each dataset in subplot (c) is generated by an adversarial data agent after each consecutive loss back-propagation.

TabPFN and remained within a second on GPU (NVIDIA H100), showing that neural modifications from the mixture block have not made APT significantly heavier. Note that there is no cherry-picking being performed on model checkpoints for APT – the APT model that we released and used for evaluations is the last model after the final iteration of pre-training. Realistically, PFN-based models are pre-trained on synthetic data, and picking checkpoints for evaluations ad hoc is not ideal unless using a whole different collection of real-world datasets for validation. But even in that case, it would still raise the concern of data leakage.

In these experiments, the deep learning algorithms under the standard supervised learning pipeline, ResNet and SAINT, yielded subpar performances. Note that the computing budget in Hollmann et al. (2022) and ours is set to 1 hour per dataset per split contrary to the 10 hours in McElfresh et al. (2024). The deep learning algorithms under the zero-shot meta-learning pipeline, APT and TabPFN, yielded ideal performances, but it has been previously shown that TabPFN sees a significant drop in performance on datasets with categorical features or missing values (Hollmann et al., 2022). In Figure 4, we further break down the results on datasets with and without these characteristics.



*Figure 4.* A breakdown of performance by dataset characteristics. The mean of ranks are plotted as orange on datasets with the respective characteristic, and as blue on datasets without the respective characteristic.

From Figure 4, it can be observed that APT has fairly dealt with TabPFN's weakness in handling datasets with missing values, and has closed the gap between performance on datasets with and without categorical features compared to TabPFN, although GBDTs such as CatBoost still shows the greatest capability in handling datasets with categorical features. We further break down the performance contributions from each proposed component in Section 3.4.

### 3.2. Qualitative Analysis of the Adversarial Data Agents

Even though arbitrary MLPs have the potential to serve as universal function approximators given certain regularity constraints (Hornik et al., 1989), the pre-set hyperparameters (e.g. sampling distribution of neurons, sampling distribution of the number of layers, choices of activations, etc.) as well as the lack of gradient updates restrict the family of data distributions that randomly initialized sparse neural networks can put forward in practice. As shown in Figure 3, the distribution of two-dimensional data generated by two whole different sets of random sparse neural networks align fairly precisely with merely 2,000 independent initializations. On the contrary, even without resetting neural architecture and neural parameters, the adversarial data agents still managed to generate a more diverse collection of data and diffuse the concentrated peaks presented in the density distribution of data generated by ordinary data generators. To be exact, for a collection of $2,000$ datasets generated by ordinary data generators, we evaluated a KL-divergence of $0.134 \pm 0.141$ between it and a collection of $2,000$ datasets generated by another set of ordinary data generators, and a KL-divergence of $0.813 \pm 0.072$ between it and a collection of $2,000$ datasets generated by adversarial data agents.

As a motivation of imposing data agent reset, we were wary that the data agents after many adversarial updates could yield synthetic datasets whose features have little to no

signal on the response variable. With our hyperparameter settings, we have not observed such behavior and to our surprise, the synthetic datasets generated by adversarial agents exhibit slightly stronger signal with a Pearson correlation of $0.311 \pm 0.026$ between predictors and responses on datasets with two-dimensional features as oppose to the $0.268 \pm 0.013$ of ordinary data generators. We postulate that this is partially in consequence of the high reset frequency and high generator learning rate.

### 3.3. Generalizability of the Mixture Block

After a ZSML is deployed, one should not be required to re-do its pre-training given certain characteristics of the datasets in evaluation pool that the model cannot handle, and this is why the mixture block architecture is important. For TabPFN, we have to look at the evaluation dataset pool first, calculate the largest class size, before using it as a hyperparameter for pre-training. This is not a procedure that fits well into the zero-shot learning concept. Our proposed mixture block architecture does not have such class size limitation, and we show the performance of APT on datasets with more than 10 classes in OpenML-CC18 in Table 2.

*Table 2.* The ROC-AUC on datasets with more than 10 classes. APT pre-trained on datasets with a maximum of 10 classes is able to match APT without mixture block pre-trained on datasets with a maximum of 26 classes on 3 of the 4 datasets.

|  | letter | isolet | vowel | texture |
|---|---|---|---|---|
| APT w/o Mixture | $.975 \pm .002$ | $.970 \pm .003$ | $1 \pm 0$ | $1 \pm 0$ |
| APT | $.975 \pm .002$ | $.939 \pm .011$ | $1 \pm 0$ | $1 \pm 0$ |

Interestingly, the mixture block's generalizability significantly accelerated pre-training in our experiments. The ROC-AUC evaluated after each iteration of pre-training with and without the mixture block is presented in Figure 5. Note that ensembling over permutations (Hollmann et al., 2022) is not performed in this experiment as it would dramatically increase runtime given that evaluation is performed following every gradient step.

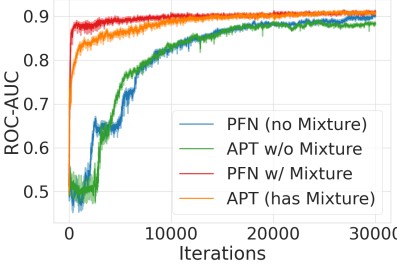

*Figure 5.* ROC-AUC on the 35 small datasets in OpenML-CC18 evaluated after each of the first 30,000 gradient steps.

From Figure 5, we can see that models with mixture block

learn remarkably faster than models without it. For APT, the model reaches an AUC of $0.70$ in merely 40 gradient steps, an AUC of $0.80$ in 591 gradient steps and $0.90$ in 11,780 gradient steps.

### 3.4. Ablation Study

**Classification** Although we discovered that the mixture block gives the model a nice performance acceleration in the previous section, the original purpose of designing such architecture was not performance-driven, and we still expect that the final performance improvement was largely contributed by the adversarial pre-training. We present ablation study in Table 3 to verify this expectation.

*Table 3.* Ablation study on tabular classification. Note that APT is TabPFN with adversarial pre-training and mixture block.

|  | Small | | All | |
|---|---|---|---|---|
|  | mean AUC ↑ | rank ↓ | mean AUC ↑ | rank ↓ |
| APT | $\mathbf{0.921} \pm 0.003$ | $2.11 \pm 0.16$ | $\mathbf{0.918} \pm 0.006$ | $\mathbf{2.1} \pm 0.2$ |
| APT w/o Mixture | $0.917 \pm 0.005$ | $\mathbf{2.09} \pm 0.06$ | $0.917 \pm 0.005$ | $\mathbf{2.1} \pm 0.1$ |
| TabPFN w/ Mixture | $0.914 \pm 0.004$ | $2.55 \pm 0.22$ | $0.914 \pm 0.005$ | $2.6 \pm 0.2$ |
| TabPFN | $0.913 \pm 0.003$ | $2.49 \pm 0.16$ | $0.914 \pm 0.005$ | $2.4 \pm 0.2$ |

Unsurprisingly, models with and without the mixture block did not dominate each other on mean AUC and rank collectively. Note that the mixture block was proposed for generalizing on datasets with unseen number of classes, and we expect it to have little to no impact on datasets with seen number of classes performance-wise.

**Regression** Although ZSMLs are gradually catching up with GBDTs on classification problems and likely reached a performance mark close to saturation on small classification problems, tabular regression remains an area where ZSMLs have not yet shown exceptional performance. We additionally report a study on the 35 datasets in OpenML-CTR23 regression suite in Table 4, and show the progress APT has made on regression tasks over TabPFN.

*Table 4.* Ablation study on tabular regression. Small datasets are the 12 datasets in OpenML-CTR23 with data size no larger than 2,000. Note that APT is TabPFN with adversarial pre-training in this setting, since the mixture block was only used for classification tasks.

|  | Small | | All | |
|---|---|---|---|---|
|  | mean MSE ↓ | wins ↑ | mean MSE ↓ | wins ↑ |
| TabPFN | $0.412 \pm 0.077$ | $3.8 \pm 1.2$ | $0.340 \pm 0.025$ | $6.4 \pm 1.4$ |
| APT | $\mathbf{0.344} \pm 0.068$ | $\mathbf{8.2} \pm 1.2$ | $\mathbf{0.306} \pm 0.023$ | $\mathbf{28.6} \pm 1.4$ |

From Table 4, it can be observed that incorporating adversarial pre-training has boosted the performance of TabPFN, yielding a larger number of wins with a significant margin. Note that we used the exact same synthetic data sampling distributions and hyperparameters that were used in TabPFN

for the purpose of ablation, in order to clearly demonstrate the contribution of adversarial training. TabPFN was trained only on classification problems, and therefore it is possible that these hyperparameters are over-optimized for classification tasks, and under-optimized for regression tasks.

# 4. Related Work

## 4.1. Tabular Learning

GBDTs such as XGBoost and others (Chen & Guestrin, 2016; Prokhorenkova et al., 2018; Ke et al., 2017) are commonly used for tabular data problems, in the traditional one-model-for-one-dataset approach. At this point, numerous deep learning approaches have been developed for tabular data, mostly taking the one-model-for-one-dataset approach (Borisov et al., 2022; Somepalli et al., 2021; Gorishniy et al., 2021; Rubachev et al., 2022; Kadra et al., 2021a; Arik & Pfister, 2021; Popov et al., 2019; Arik & Pfister, 2021; Kotelnikov et al., 2023; Gorishniy et al., 2024; 2022; Chen et al., 2024; Kadra et al., 2021b; Huang et al., 2020), but some also venturing into transfer learning, many but not all leveraging large language models to find relevant information for the tabular data problem at hand (Levin et al., 2022; Yan et al., 2024; Borisov et al., 2023; Ye et al., 2024a; Spinaci et al., 2024; Hegselmann et al., 2023; Kim et al., 2024; Zhu et al., 2023).

**Tabular Meta-Learning** Auto-Sklearn introduced in Feurer et al. (2015) and improved upon in Feurer et al. (2022) use Bayesian optimization to determine the best algorithm and feature pre-processing steps for modeling a given dataset. Meta learning is used for initializing the Bayesian optimization. In contrast to Auto-Sklearn and methods of transfer learning for tabular data, TabPFN (Müller et al., 2021) is trained solely on synthetic data to learn the general prediction logic of tabular classification and to acquire meaningful data representations in inference time. Helli et al. (2024) introduced a variant of TabPFN that was trained on a drifting synthetic data distribution, but the drift is independent of the performance of the model being optimized.

## 4.2. Zero-shot Learning

Recent work such as Xian et al. (2018; 2017); Chang et al. (2008); Larochelle et al. (2008); Palatucci et al. (2009) have shown impressive capability of zero-shot learning in the space of language and vision problems. Recent approaches to zero-shot or few-shot learning for tabular data problems mostly encode tabular data as language, and then leverage large language models (LLMs) for their zero- or few-shot capabilities (see Hegselmann et al. (2023); Nam et al. (2023); Gardner et al. (2024)). These approaches rely on relevant information about the tabular data existing in LLMs – this is most obviously the case when column names are mean-

ingful, but not guaranteed for broad tabular data problems.

## 4.3. Adversarial Training

Upon generative adversarial networks (GANs) (Goodfellow et al., 2015; Madry et al., 2018; Kurakin et al., 2017), recent work such as Shafahi et al. (2019) improved on the efficiency by combining the back-propagation steps of the generator and discriminator. However, this method has been shown to suffer from catastrophic overfitting (Andriushchenko & Flammarion, 2020; Kim et al., 2021) without further modifications. Other works focusing on improving the efficiency of GAN training include Wong et al. (2020) and Zhang et al. (2019) where they restrict most of the forward and back propagation within the first layer of the network during adversarial updates. Zhang et al. (2021) in particular noted that weight updates frequently go back and forth in opposite directions in one training epoch, suggesting those updates might be redundant. Many other variations have been introduced to mitigate vanishing gradient and additional challenges of GAN training (Jabbar et al., 2021): failing at finding a Nash-equilibrium (Ratliff et al., 2016), and internal covariate shift (Ioffe, 2015).

# 5. Conclusion

In this work, we gave the first effort in exploring the adversarial pre-training of deep zero-shot meta-learners, specifically PFNs. We proposed APT, a zero-shot meta-learner that improves the performance of TabPFN on tabular prediction tasks and matches state-of-the-art GBDTs on small tabular classification tasks. In addition, we proposed a mixture block neural design to eliminate the class size restriction of PFNs, addressing a crucial issue in their generalizability to broad classification problems. As for limitations, APT does not outperform GBDTs on large tabular datasets, and shares the quadratic computational complexity of TabPFN. Hence, extensions of this work could explore means of acquiring data representations in a more inexpensive manner. For example, considerable research in recent years has significantly accelerated the transformer and increased its context length (Wu et al., 2022; Bulatov et al., 2023). It is a worthwhile effort for future research to apply these advancements to APT as well as other PFNs. Besides, future research could extend the mixture block to standard (non-zero-shot) classification settings in light of its ability to generalize and greatly accelerate convergence, which could improve the performance of traditional DNNs on small classification datasets. Mixture block or other alternatives to the dense final layer could also be explored in both standard and zero-shot regression settings, which could have an impact on the inductive bias of DNNs and their underperformance in comparison to GBDTs (Grinsztajn et al., 2022) under certain tabular data nature.

## Acknowledgements

We thank Tyler Farnan, Gang Mei, and C. Bayan Bruss for the insightful discussions.

## Impact Statement

This paper presents work whose goal is to advance the field of Machine Learning. There are many potential societal consequences of our work, none which we feel must be specifically highlighted here.

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

# A. Background

In this section, we give a brief introduction to PFNs, and specifically the synthetic data generating mechanism of TabPFN. For a more complete description, see Müller et al. (2021); Hollmann et al. (2022); Nagler (2023). Given training dataset $D^{\text{train}} = (X^{\text{train}}, \boldsymbol{y}^{\text{train}})$, the goal is to approximate the conditional outcome distribution $y^{\text{test}} \sim p(\cdot | \boldsymbol{x}^{\text{test}}, D^{\text{train}})$ given a test point $\boldsymbol{x}^{\text{test}}$. In the Bayesian framework for supervised learning, the prior of the dataset is a hypothesis of the data generating mechanism $\phi$ drawn from hypothesis space $\Phi$, under which $p(\cdot | \boldsymbol{x}^{\text{test}}, D^{\text{train}})$ is a posterior predictive distribution (PPD) and can be factorized as follows by the Bayes' rule:

$$p(\cdot | \boldsymbol{x}^{\text{test}}, D^{\text{train}}) = \int_{\phi \in \Phi} p(\cdot | \boldsymbol{x}^{\text{test}}, \phi) p(\phi | D^{\text{train}}) \, d\phi \tag{3}$$

$$= \int_{\phi \in \Phi} p(\cdot | \boldsymbol{x}^{\text{test}}, \phi) \frac{p(\phi) p(D^{\text{train}} | \phi)}{p(D^{\text{train}})} \, d\phi \tag{4}$$

$$\propto \int_{\phi \in \Phi} p(\cdot | \boldsymbol{x}^{\text{test}}, \phi) p(D^{\text{train}} | \phi) p(\phi) \, d\phi. \tag{5}$$

PFNs conduct synthetic prior fitting by defining a family of data generating mechanisms $\Phi$ from which independent samples $\boldsymbol{x}_i \sim p(\boldsymbol{x}_i) = \mathbb{E}_{p(\phi)}[p(\boldsymbol{x}_i | \phi)]$ and $y_i \sim p(y_i) = \mathbb{E}_{p(\phi)}[p(y_i | \boldsymbol{x}_i, \phi)]$ are drawn to compose feature matrix $(X^{\text{train}}, X^{\text{test}})$ and response vector $(\boldsymbol{y}^{\text{train}}, \boldsymbol{y}^{\text{test}})$ of a synthetic dataset $D = D^{\text{train}} \cup D^{\text{test}}$, and use a transformer model $q_\theta(\cdot | X^{\text{test}}, D^{\text{train}})$ to approximate $p(\cdot | X^{\text{test}}, D^{\text{train}})$ by minimizing their expected divergence over the synthetic data distribution

$$\mathbb{E}_{p(D^{\text{train}}, X^{\text{test}})} \left[ \text{KL} \left( p(\boldsymbol{y}^{\text{test}} | X^{\text{test}}, D^{\text{train}}), q_\theta(\boldsymbol{y}^{\text{test}} | X^{\text{test}}, D^{\text{train}}) \right) \right]. \tag{6}$$

Since

$$\nabla_\theta \mathbb{E}_{p(D^{\text{train}}, X^{\text{test}})} \left[ \text{KL} \left( p(\boldsymbol{y}^{\text{test}} | X^{\text{test}}, D^{\text{train}}), q_\theta(\boldsymbol{y}^{\text{test}} | X^{\text{test}}, D^{\text{train}}) \right) \right] \tag{7}$$

$$= \nabla_\theta \mathbb{E}_{p(D^{\text{train}}, X^{\text{test}})} \left[ \text{H} \left( p(\boldsymbol{y}^{\text{test}} | X^{\text{test}}, D^{\text{train}}), q_\theta(\boldsymbol{y}^{\text{test}} | X^{\text{test}}, D^{\text{train}}) \right) \right] \tag{8}$$

$$= \nabla_\theta \mathbb{E}_{p(D^{\text{train}}, D^{\text{test}})} \left[ -\log q_\theta(\boldsymbol{y}^{\text{test}} | X^{\text{test}}, D^{\text{train}}) \right], \tag{9}$$

it is equivalent to minimizing the expected negative log-likelihood loss

$$\mathcal{L}(q_\theta) = \mathbb{E}_{p(D)} \left[ -\log q_\theta(\boldsymbol{y}^{\text{test}} | X^{\text{test}}, D^{\text{train}}) \right]. \tag{10}$$

TabPFN in particular, conducts synthetic prior fitting by defining a family of sparsified-random-MLP-based data generating mechanisms $\Phi$, which we call ordinary data generators in the context of this paper. The following section gives a detailed description of the workflow of these generators.

## A.1. Ordinary Data Generator

To sample data generating mechanism $\phi \sim \Phi$, TabPFN first initializes a random MLP by sampling a collection of hyperparameters from a pre-defined hyperparameter space, including number of layers, hidden size, activation function, dropout probability, noise scales, etc. Specifically, dropout probability is used to sparsify neural connections between neurons, and noise scales dictate the amount of random noise injected into neurons at each layer. After the sparsified noisy random MLP is initialized, TabPFN randomly selects a subset of neurons in this MLP to be predictors $\boldsymbol{x}_i$, and randomly select one neuron to be response $y_i$. With $n$ different random inputs to the MLP, a dataset with $n$ instances of $(x, y)$ is thus generated.

**Discretization** Since generated data are selected neurons from MLPs, their values are naturally continuous. To mimic real-world datasets that possess categorical features and to generate discrete class labels for classification tasks, TabPFN uses a ranking discretization approach that converts a subset of continuous values to discrete by designating certain quantile ranges of the continuous value $v$ to certain categories. A visual demonstration of this conversion can be found on the left side of Figure 6.

**Normalization** The generated synthetic data (as well as real-world datasets during inference time) are normalized across samples within each dataset, with the range of the values clipped to four standard deviations. Although the meta-learner might see datasets with unseen fields and out-of-distribution predictor-response relations during inference time, this ensures that at least the range of values will not be out-of-distribution as well.

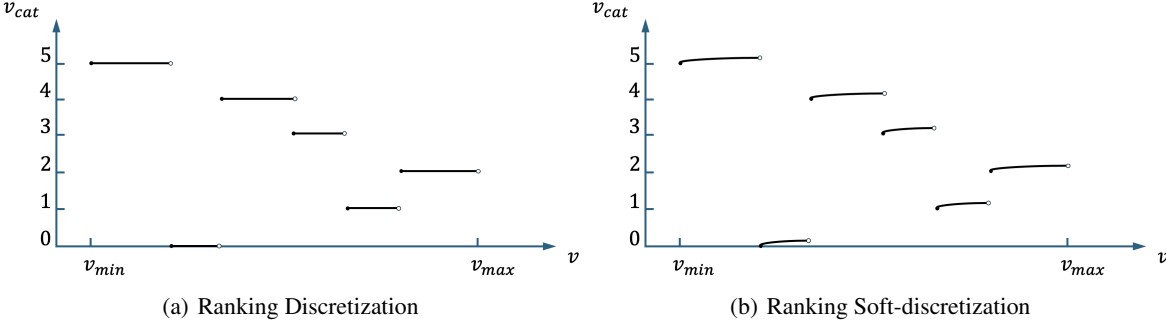

(a) Ranking Discretization

(b) Ranking Soft-discretization

*Figure 6.* Discretization of continuous variables. $x$-axis is the value generated by the data generator, and $y$-axis is its value after discretization. The soft-discretization approach produces near-categorical features that are differentiable and thus do not disrupt gradient flow. Intuitively, the adversarial data agents will try to produce new value that escapes the range of the current category if the meta-learner becomes very good at identifying signal from the current category. However, the new category it escapes to is arbitrary and cannot be targeted by gradient updates, giving additional exploration potentials to the adversarial agents.

### A.2. Limitations

Although there is no theoretical limitation on the number of data PFNs can handle, the transformer architecture does entail significant computation complexity and memory usage for large datasets. Besides, given the nature of dense input embedding layer and dense final prediction layer, there is a theoretical limitation on the number of features and the number of classes that PFNs can handle. The former is less of an issue since feature selections or simply random sampling of features can be performed, and PFNs would still yield ideal performance as shown in McElfresh et al. 2024. The latter is a rather big problem for classification tasks because there is hardly any direct and effective work-around.

## B. Handling of Large Datasets

Since expanding TabPFN's capabilities in handling large datasets is not the focus of this work, we only provide two simple adaptations such that APT can practically handle datasets of this nature. We recommend that practitioners try out the concurrent and future developments in more involved model innovations for large datasets, but in case they do not wish to do so, the following approaches can serve as a baseline.

### B.1. Uncertainty-based Batch Aggregation

For datasets with large number of samples, to avoid calculating attention spanning all training data points which results in quadratic order of operations and memory usage with respect to data size, we estimate the PPD with batches drawn from the training set:

$$p(\boldsymbol{y}^{\text{test}}|X^{\text{test}}, D^{\text{train}}) \approx \int_b q_\theta(\boldsymbol{y}^{\text{test}}|X^{\text{test}}, b) \cdot p(b|D^{\text{train}}), \tag{11}$$

which is equivalent to drawing uniform samples from training set $D^{\text{train}}$ and scale the resulting predictions with weights $p(b|D^{\text{train}})$. We cap the batch size at 3000 in alignment with McElfresh et al. (2024). For classification datasets with number of samples larger than 3000, we split the training set into batches and weigh the resulting predictions in proportion to the batch size (prediction from the last batch may have less weight than the others). For regression tasks, we parameterize model $q_\theta(y_i^{\text{test}}|\boldsymbol{x}_i^{\text{test}}, b)$ as Gaussian distribution $(\mu_\theta(\boldsymbol{x}_i^{\text{test}}, b), \sigma_\theta(\boldsymbol{x}_i^{\text{test}}, b))$ similar to Hollmann et al. (2022), and directly produce the point estimation using the inverse variance estimator in inference time:

$$\mathbb{E}[y_i^{\text{test}}|\boldsymbol{x}_i^{\text{test}}, D^{\text{train}}] \approx \int_b \mathbb{E}_\theta[y_i^{\text{test}}|\boldsymbol{x}_i^{\text{test}}, b] \cdot p(b|D^{\text{train}}) \tag{12}$$

$$= \left( \sum_k \frac{1}{\sigma_\theta^2(\boldsymbol{x}_i^{\text{test}}, b_k)} \right)^{-1} \sum_{k=1}^{N} \frac{1}{\sigma_\theta^2(\boldsymbol{x}_i^{\text{test}}, b_k)} \mu_\theta(\boldsymbol{x}_i^{\text{test}}, b_k). \tag{13}$$

The intuition is that, prediction on each batch is weighted by its uncertainty – more weights are put to the predictions that the model is more certain of, and vice versa.

### B.2. Patch-based Feature Embedding

We drew inspiration from Dosovitskiy et al. (2020) and developed a patch-based embedding approach that adapts to datasets with arbitrary number of features. In Hollmann et al. (2022), embeddings of $x$ are acquired by padding or clipping the number of features $d_k$ to a certain maximum feature size $d^*$, such that $x$ can be fed to a dense feedforward $e_\theta : \mathbb{R}^{d^*} \to \mathbb{R}^{d_{\text{model}}}$. Instead, we split features into patches, setting $d^*$ as the patch size, and only pad the last patch to $d^*$ dimensions if $d \not\equiv 0$ $(\text{mod } d^*)$. After feeding each patch to dense feedforward $e_\theta$, we pass them to an attention block with optional relative positional encoding (Su et al., 2021; Press et al., 2021), and average pool across the resulting embeddings of patches. Essentially, this is a half-way approach between using a dense feedforward to embed all features, and using an attention block to tokenize each individual feature. In this way, the embedding block can handle features in a more flexible manner while controlling computational complexity and memory usage.

## C. Hyperparameter Settings

All common hyperparameters of APT are directly inherited from TabPFN and not tuned, including learning rate $10^{-4}$, number of blocks $12$, hidden dimensions $512$, hidden feedforward dimensions $1024$, number of heads $4$, effective batch size (batch size per step $\times$ number of gradient accumulation steps) $64$, total number of training datasets (number of epochs $\times$ steps per epoch $\times$ number of datasets per step) $6,400,000$, as well as all data generator hyperparameters. For more details on the data generator hyperparameters, see the code repository in our supplementary material.

## D. More Results

We list the performance of top algorithms on small classification datasets in Table 5. Standard deviations are calculated across 5 different splits.

Table 5. The ROC-AUC of top algorithms on the 35 small datasets in OpenML-CC18.

| | LightGBM | XGBoost | SVM | TabPFN | CatBoost | APT |
|---|---|---|---|---|---|---|
| mfeat-fourier | .981 ± .004 | .982 ± .004 | .982 ± .004 | .985 ± .002 | .984 ± .002 | .983 ± .003 |
| breast-w | .993 ± .006 | .993 ± .006 | .995 ± .007 | .997 ± .003 | .996 ± .005 | .997 ± .003 |
| mfeat-karhunen | .999 ± .001 | .999 ± .001 | 1 ± 0 | .999 ± 0 | .999 ± 0 | 1 ± 0 |
| mfeat-morphological | .959 ± .004 | .961 ± .002 | .965 ± .006 | .967 ± .003 | .964 ± .003 | .966 ± .006 |
| mfeat-zernike | .970 ± .004 | .973 ± .004 | .992 ± .003 | .982 ± .001 | .974 ± .003 | .977 ± .003 |
| cmc | .751 ± .036 | .758 ± .036 | .690 ± .020 | .736 ± .031 | .758 ± .037 | .739 ± .026 |
| credit-approval | .931 ± .030 | .920 ± .022 | .912 ± .024 | .928 ± .029 | .931 ± .030 | .930 ± .022 |
| credit-g | .809 ± .018 | .824 ± .028 | .816 ± .020 | .835 ± .018 | .816 ± .025 | .846 ± .024 |
| diabetes | .821 ± .027 | .812 ± .037 | .811 ± .050 | .817 ± .026 | .827 ± .025 | .824 ± .016 |
| tic-tac-toe | 1 ± 0 | 1 ± 0 | 1 ± 0 | .993 ± .003 | 1 ± 0 | .997 ± .002 |
| vehicle | .936 ± .009 | .945 ± .008 | .965 ± .011 | .965 ± .005 | .941 ± .008 | .961 ± .008 |
| eucalyptus | .900 ± .022 | .894 ± .024 | .874 ± .009 | .908 ± .013 | .905 ± .019 | .912 ± .017 |
| analcatdata_authorship | 1 ± 0 | 1 ± 0 | 1 ± 0 | 1 ± 0 | 1 ± 0 | 1 ± 0 |
| pc4 | .953 ± .008 | .954 ± .012 | .907 ± .058 | .957 ± .013 | .961 ± .011 | .964 ± .016 |
| pc3 | .814 ± .031 | .831 ± .048 | .706 ± .055 | .848 ± .044 | .829 ± .042 | .865 ± .032 |
| kc2 | .887 ± .060 | .862 ± .102 | .881 ± .052 | .875 ± .079 | .885 ± .084 | .896 ± .087 |
| blood-transfusion-service-center | .740 ± .085 | .722 ± .068 | .705 ± .075 | .750 ± .082 | .732 ± .077 | .751 ± .086 |
| cnae-9 | .981 ± .005 | .994 ± .005 | .998 ± .001 | .812 ± .032 | .991 ± .005 | .901 ± .014 |
| ilpd | .767 ± .067 | .751 ± .038 | .628 ± .085 | .792 ± .046 | .787 ± .059 | .808 ± .035 |
| wdbc | .993 ± .006 | .989 ± .007 | .998 ± .003 | .997 ± .003 | .993 ± .003 | .997 ± .004 |
| dresses-sales | .685 ± .028 | .618 ± .045 | .669 ± .027 | .552 ± .056 | .637 ± .051 | .617 ± .049 |
| MiceProtein | 1 ± 0 | 1 ± 0 | 1 ± 0 | 1 ± 0 | 1 ± 0 | 1 ± 0 |
| steel-plates-fault | .975 ± .003 | .979 ± .003 | .964 ± .006 | .970 ± .005 | .978 ± .003 | .969 ± .006 |
| climate-model-simulation-crashes | .944 ± .043 | .936 ± .052 | .951 ± .070 | .960 ± .053 | .949 ± .044 | .960 ± .058 |
| balance-scale | .970 ± .027 | .998 ± .003 | .994 ± .006 | .997 ± .004 | .949 ± .014 | .998 ± .003 |
| mfeat-factors | .999 ± .001 | .999 ± .001 | .999 ± .001 | .999 ± .001 | .999 ± 0 | .999 ± .001 |
| vowel | .999 ± .001 | .999 ± .001 | .999 ± .001 | 1 ± 0 | 1 ± 0 | 1 ± 0 |
| analcatdata_dmft | .595 ± .032 | .597 ± .029 | .601 ± .033 | .577 ± .044 | .582 ± .027 | .593 ± .040 |
| pc1 | .901 ± .065 | .917 ± .063 | .802 ± .127 | .917 ± .059 | .916 ± .058 | .942 ± .041 |
| banknote-authentication | 1 ± 0 | 1 ± 0 | 1 ± 0 | 1 ± 0 | 1 ± 0 | 1 ± 0 |
| qsar-biodeg | .934 ± .015 | .925 ± .012 | .932 ± .017 | .944 ± .016 | .935 ± .017 | .944 ± .013 |
| semeion | .998 ± .001 | .999 ± .001 | .999 ± 0 | .984 ± .004 | .999 ± .001 | .980 ± .004 |
| cylinder-bands | .898 ± .041 | .873 ± .036 | .913 ± .035 | .911 ± .021 | .904 ± .044 | .913 ± .031 |
| car | 1 ± 0 | 1 ± 0 | 1 ± 0 | .999 ± .001 | 1 ± 0 | .997 ± .005 |
| mfeat-pixel | .999 ± 0 | 1 ± 0 | 1 ± 0 | .999 ± 0 | 1 ± 0 | .999 ± 0 |

