# OpenReview forum: "Zero-shot Meta-learning for Tabular Prediction Tasks with Adversarially Pre-trained Transformer"
_ICML.cc/2025/Conference — ICML 2025 poster_

### Official Review · Reviewer_TeqX · 2025-03-12

**Overall Recommendation:** 2

**Summary:**

This paper introduces APT, extending on PFNs and TabPFN, which leverages adversarial synthetic data agents for pretraining and incorporates a mixture block architecture to handle classification tasks with an arbitrary number of classes, addressing the class size limitation.

**Claims And Evidence:**

No, the authors claim that they employ meta-learning methods for tabular prediction in the zero-shot scenarios. However, this claim is not adequately reflected in the method section, leaving the meta-learning process and its adaptation to zero-shot scenarios unclear.

**Essential References Not Discussed:**

No

**Experimental Designs Or Analyses:**

Yes, the authors conduct the experiments that adversarial pre-training enhances the performance of TabPFN, showing that adversarial synthetic data agents generate more diverse datasets compared to standard random generators in TabPFN. However, this paper does not clearly explain how APT effectively performs meta-learning or adapts to zero-shot scenarios.
Additionally, while mixture block architecture design has been shown to improve generalization and significantly accelerate pretraining, Table 1 indicates that most selected baselines are not designed for the zero-shot scenarios. This raises concerns about the effectiveness of APT in truly zero-shot settings.

**Methods And Evaluation Criteria:**

Yes

**Other Comments Or Suggestions:**

We suggest that the author should provide a detailed definition of zero-shot meta-learning for tabular prediction and clarify the significance of zero-shot scenarios in tabular prediction tasks, specifically explaining why zero-shot table prediction is necessary.

**Other Strengths And Weaknesses:**

Strengths:
1. Data Agent Reset helps improve the diversity of generated data.

Weaknesses:
1. APT extended on PFN and TabPFN, which is proposed to perform zero-shot meta-learning on tabular prediction tasks without using any real-world dataset to pre-train the model, but the authors do not clearly explain its meta-learning mechanism and how it enables zero-shot adaptation in the task of tabular prediction.
2. The research motivation of this paper is not clear, authors do not clarify the significance of zero-shot table prediction or its practical implications. Additionally, authors do not also provide a clear task definition for zero-shot meta-learning in tabular prediction.
3. The experimental baselines primarily consist of some methods with requiring labeled samples for effective training. This raises concerns about whether directly comparing these methods with APT in a zero-shot scenario meaningfully reflects APT’s zero-shot capabilities.
4. As shown in the Table1, APT does not exhibit a significant improvement in ROC-AUC when compared to CatBoost and TabPFN.

**Questions For Authors:**

1. How does APT perform meta-learning, and how does it enable zero-shot adaptation?
2. Are the selected baseline methods appropriate for evaluating APT’s zero-shot capabilities, given that they require labeled samples for training?
3. What are the advantages of APT compared to CatBoost and TabPFN?

**Relation To Broader Scientific Literature:**

This work extends on TabPFN by enhancing the diversity of its pretraining dataset through adversarial generation and enabling arbitrary classification through a mixture block model architecture. TabPFN, leveraging context learning and PFN, has demonstrated strong performance in small-scale tabular prediction.

**Theoretical Claims:**

The paper does not introduce any significant theoretical contributions on meta learning in the zero-shot scenarios.

---

> ### Author Rebuttal · Authors · 2025-04-01
>
> We thank reviewers for their valuable feedback and comments. We see that the reviewer’s main concern lies in the confusion about zero-shot meta-learning, so first and foremost, we want to address this concern and try our best to provide the reviewer a clear picture.
>
> ---
>
> Weakness 1 & 2 (Question 1):
>
> **Response**: As the reviewer has noticed, APT only improves on TabPFN’s pre-training procedure and inference-time generalizability, and that’s the goal of this research. While we believe these innovations are interesting and significant, they do not change the fundamental Bayesian mechanism of TabPFN as well as how and why it can perform meta-learning. TabPFN is a zero-shot meta-learner. We will reiterate the precise definition in the next paragraph, but for a detailed explanation on how and why PFN-based models can perform meta-learning, how and why PFN-based models can adapt to zero-shot scenarios, and why zero-shot tabular prediction is important, please kindly see [1] and [2]. Per the reviewer’s feedback, we have added a sentence at L75-76 left column in the introduction section that reads *“PFN-based methods, such as TabPFN, are zero-shot meta-learners. For more details on how PFN’s underlying Bayesian mechanism can perform meta-learning with zero optimization on an unseen dataset, see [1].”* to clarify that PFN-based models' capabilitiy to perform zero-shot meta-learning is *not* where our unqiue contribution lies.
>
> Some recent TabPFN-based works tend to call this class of method “tabular foundation models” [3][4], while the term tabular “zero-shot meta-learner” is slightly more specific. We have provided the definition of zero-shot meta-learner in the introduction section, and we will further highlight it in the proposed method section per reviewer’s suggestion:
>
> - A model that performs no optimization (i.e. zero gradient updates for a deep learning model) on its parameters while predicting on unseen datasets.
>
> This is mentioned in [5] and it is a simple extension on the definition of meta-learner:
>
> - A model that performs few optimizations (i.e. few gradient updates for a deep learning model) on its parameters while predicting on unseen datasets.
>
> We chose to not categorize PFN-based methods as foundation models because the reason why PFN-based models work is very different from the reason why large vision and language models work – they do not pretrain on any real-world data and their goal is to learn to adaptively acquire data representation of unseen data during inference time. Nonetheless, both concepts have very general and straightforward definitions, and are both correct categorizations of PFNs. We hope that this choice of categorization would not be a major deciding factor, as how PFN-based models perform meta-learning under the zero-optimization scenario is well documented in [1].
>
> ---
>
> Weakness 3 (Question 2):
>
> **Response**: Yes, this baseline selection is appropriate. It is no different from the baseline selection of of other prior works that benchmark PFNs such as [1], TabPFN, and TabZilla [6], as Reviewer pxwT also pointed out. We want to respectfully note that the reviewer might have a slight misunderstanding regarding the evaluation setups of PFN-based works? (If this is not the case, please kindly ignore this comment) The traditional methods such as CatBoost in the evaluations are not trained under zero-shot scenarios, they are trained on the full training sets and given much more computational time for tuning – this actually gives zero-shot methods more disadvantages and further showcases their zero-shot capabilities. The research on tabular zero-shot meta-learning is still very young, and there are no alternatives to select as baselines other than PFNs, to the best of our knowledge.
>
> ---
>
> Weakness 4 (Question 3):
>
> **Response**: The advantage of APT compared to TabPFN is its inference-time generalizability (it can predict on datasets with unseen classes of unseen cardinality) and pre-training data diversity; the advantage of APT compared to CatBoost is its exceptional runtime thanks to the zero-shot capability. We want to note that the sheer performance on small tabular classification tasks is nearly saturated [6], and none of the recent zero-shot meta-learners or other deep methods is able to significantly beat GBDTs such as CatBoost on small tabular prediction tasks with no limitation on feature size, class size, or number of missing values [1][6]. We believe the data diversity improvement in pre-training as well as the proposal of the mixture block that fundamentally addressed TabPFN’s class size limitation are important contributions to PFNs aside from the improved performances, and we believe we have showcased their power through extensive ablations.
>
> ---
>
> *References*:
>
> [1] [S Müller, et al. ICLR 2022]
>
> [2] [T Nagler. ICML 2023]
>
> [3] [N Hollmann, et al. Nature 2025]
>
> [4] [SB Hoo, et al., Arxiv 2501.02945]
>
> [5] [VK Verma, et al. AAAI 2020]
>
> [6] [D McElfresh, et al. NeurIPS 2024]

---

> > ### Comment · Reviewer_TeqX · 2025-04-07
> >
> > I confirm that I have read the author response to my review and will update my review in light of this response as necessary.

---

### Official Review · Reviewer_pxwT · 2025-03-13

**Overall Recommendation:** 3

**Summary:**

This work introduces APT, a zero-shot meta-learning model for tabular prediction, pre-trained with adversarial synthetic data agents. It improves TabPFN, removes class size limitations via a mixture block architecture, and matches SOTA GBDTs on small tabular tasks. While enhancing performance in classification and regression, APT retains quadratic scaling and struggles on large datasets.

**Claims And Evidence:**

Yes, the claims are supported by clear and convincing evidence.

**Essential References Not Discussed:**

No.

**Experimental Designs Or Analyses:**

Yes

**Methods And Evaluation Criteria:**

Yes, it follows the previous works of tabpfn and tabular benchmark.

**Other Comments Or Suggestions:**

More related works of tabpfn [ICLR'23, Nature'25] should be discussed, both of them are the stem of the tabpfn. Furthermore, the tabpfn [ICLR'23] has been extended to the journal version [Nature'25] and release the tabpfn v2, more discussion about the extension and the comparison with the tabpfn v2 is needed.

**Other Strengths And Weaknesses:**

The idea of adversarially generating synthetic data is interesting.

**Questions For Authors:**

1. Recently, the extended version of tabpfn [Nature'25] is published and release the tabpfn v2, the experiments of this paper can be compared with the tabpfn v2.
2. The tabpfn [ICLR'23] use the structure causal model to generate the synthetic data, can the proposed method be applied to the structure causal model? (ref: TabPFN: A Transformer That Solves Small Tabular Classification Problems in a Second)

**Relation To Broader Scientific Literature:**

The ability of tabpfn is highly dependant on the quality of the synthetic data, this paper is trying to improve the quality of the synthetic data by leveraging the adversarial learning. It also gives a solution to remove the class size limitation of the tabpfn.

**Theoretical Claims:**

This paper does not have theoretical claims.

---

> ### Author Rebuttal · Authors · 2025-04-01
>
> We thank the reviewer for their clear and valuable feedback on our work, and we address your questions and comments as follows:
>
> ---
>
> “Other Comments Or Suggestions” & Question 1:
>
> **Response**: Thank the reviewer for the suggestion. We added citation to TabPFN v2 [Nature'25] per your comment, but we want to respectfully explain to the reviewer why comparison to TabPFN [ICLR'23] is more appropriate than comparison to TabPFN v2 under the scope of this work:
>
> - Per ICML guidelines (https://icml.cc/Conferences/2025/ReviewerInstructions), the main conference submission has the same four-month concurrent cutoff adopted from ICLR, and reviewers should not expect authors to discuss these works that have been recently made public. TabPFN v2 was made public within one month of the submission deadline.
>
> - We do not see the Nature version of TabPFN to be a better comparison to APT than the ICLR version of TabPFN. The primary reason is reproducibility. We take ablation *very* seriously in this work, and TabPFN v2 does not release their pre-training code, while the primary contribution of this work is the improvement to an existing pre-training scheme. As stated in our work, all setups and hyperparameters of our synthetic data distributions (as well as the transformer model) are set to be exactly the same as TabPFN, in order to fairly examine the impact of adversarial data agents to pre-training.
>
> ---
>
> Question 2:
>
> **Response**: Yes, the proposed method already uses the SCM-based data generating mechanism. For the purpose of ablation, we did not make any further changes to TabPFN’s underlying generating mechanism as well as hyperparameter settings, other than necessary changes that enable stable adversarial training (L304-305 left column; Appendix B; Appendix C).
>
> Note that in this work, we described the SCM-MLP simply as “sparsified noisy MLPs” to be more explicit and straightforward, since all structural equations in the so-called SCM are modeled unifiedly as linear equations plus simple activation, and the exogenous noise in the structural equations are unifiedly modeled as gaussian with the same mean and pre-sampled variance. But in light of the reviewer’s comment, we understand that it’s important to draw connections to concepts and terminologies used in prior work, so we have added this information to L137-142 left column which now reads: *“predictors $\mathbf{x}^{(k)}_i$ and response $y^{(k)}_i$ are values of randomly selected neurons in sparsified noisy MLPs (i.e. the SCM data generating mechanism) with some additional pre-processing. More details regarding this approach can be found in Appendix B.1.”*.
>
> ---
>
> We hope that in light of our responses and further clarifications, the reviewer would consider raising their score of our paper to a strong acceptance. Thank the reviewer for their time and effort.

---

### Official Review · Reviewer_WkGE · 2025-03-14

**Overall Recommendation:** 3

**Summary:**

The paper introduces an Adversarially Pre-trained Transformer (APT) for zero-shot meta-learning on tabular prediction tasks. APT is pre-trained using adversarial synthetic data agents that continuously generate challenging datasets, enabling the model to generalize to unseen tasks without requiring real-world data for training. The authors also propose a mixture block architecture to handle classification tasks with arbitrary class sizes, addressing a key limitation of prior methods like TabPFN. Experiments show that APT achieves state-of-the-art performance on small tabular classification tasks, with improved generalization and faster pre-training compared to existing methods.

## Update After Rebuttal
Thanks for the authors' response! I will keep the rating.

**Claims And Evidence:**

Yes

**Essential References Not Discussed:**

I am not in the field of tabular data paper.

**Experimental Designs Or Analyses:**

Yes

**Methods And Evaluation Criteria:**

Yes

**Other Comments Or Suggestions:**

N/A

**Other Strengths And Weaknesses:**

Strengths:
1. The use of adversarial synthetic data agents enhances the diversity and difficulty of the training data, leading to better generalization and robustness in zero-shot learning scenarios.

2. The proposed mixture block eliminates the class size limitation of previous methods, making the model more flexible and applicable to a wider range of classification tasks.

3. Extensive experiments demonstrate that APT achieves competitive or superior performance on small tabular classification tasks, demonstrating its efficiency and effectiveness.

Weakness:
1. The proposed approach performs well on small tabular datasets, but it might be difficult to be applied for large-scale dataset because of the quadratic runtime and memory complexity. A more detailed discussion on potential optimizations for scalability would strengthen the paper.
2. Regarding the improvements in regression tasks, while better than TabPFN, APT still falls short of tree-based models in regression. More discussion about this could be helpful.

**Questions For Authors:**

N/A

**Relation To Broader Scientific Literature:**

Adding synthetic adversarial training data could help the generalization of tabular model training, which can provide beneficial insights.

**Theoretical Claims:**

There's no proof in the paper

---

> ### Author Rebuttal · Authors · 2025-04-01
>
> We thank the reviewer for their succinct and extremely clear examination of our work.
>
> In light of your feedback, we have added the following paragraphs:
>
> ---
>
> Weakness 1:
>
> **Response:** *The limitations are imposed on PFNs by the transformer architecture’s quadratic computation scaling. However, considerable research in recent years has significantly accelerated the transformer and increased its context length, in some cases up to 1 million tokens [1][2]. It is a worthwhile effort for future research to apply these accelerations to PFNs beyond the scope of this paper. The architecture we employ can be modified to include these advancements.*
>
> We intend to add this paragraph to the conclusion section in the final version of the paper, together with specific references to the notable advancements in transformer capabilities. This should provide a clear roadmap for future work to improve on APT performance.
>
> *Reference*:
>
> [1] Yuhuai Wu, Markus Norman Rabe, DeLesley Hutchins, and Christian Szegedy. Memorizing transformers. ICLR 2022
>
> [2] Aydar Bulatov, Yuri Kuratov, and Mikhail S. Burtsev. Scaling transformer to 1m tokens and beyond with RMT. CoRR, 2023.
>
> ---
>
> Weakness 2:
>
> **Response:** *Note that rather than re-designing the base synthetic data sampling distributions or re-performing extensive hyperparameter search, we use the exact same synthetic data sampling distributions and hyperparameters that were used in TabPFN for the purpose of ablation, in order to clearly demonstrate the contribution of adversarial training. TabPFN was trained only on classification problems, and therefore it is possible that these hyperparameters are over optimized for classification tasks, and under optimized for regression tasks.*
>
> We will add this paragraph to end of the experiments section in the final version of the paper, to point out and attempt to explain the relative performance in regression.
>
> ---
>
> We hope that in light of including these additional brief discussions in the paper, the reviewer would consider raising their score of our paper to a strong acceptance. We appreciate the reviewer for their time and effort.

---

### Official Review · Reviewer_Ezzc · 2025-03-20

**Overall Recommendation:** 2

**Summary:**

This paper introduces the Adversarially Pre-trained Transformer (APT), which is a novel zero-shot meta-learning method for tabular data prediction tasks. By employing adversarial synthetic data agents and a mixture block architecture, APT addresses key limitations in prior tabular learning methods, particularly in handling datasets with multiple classes and complex feature distributions. APT achieves state-of-the-art performance on small tabular classification tasks, generating more diverse synthetic data while maintaining computational efficiency comparable to existing methods.

**Claims And Evidence:**

The scatter-sum mixture block represents a conceptually groundbreaking architectural approach that transcends conventional multi-layer perceptron final layer designs, offering tantalizing possibilities for cross-domain generalizability. This design is particularly compelling due to its potential to reimagine output prediction mechanisms across diverse machine learning domains.
The meticulous dataset preprocessing approach demonstrates exceptional rigor through the deliberate exclusion of four vision datasets (MNIST 784, CIFAR-10, Devnagari-Script, and Fashion-MNIST) from the original collection, ensuring a laser-focused analysis of genuinely tabular data representations.

**Essential References Not Discussed:**

It seems that there are no essential references not discussed.

**Experimental Designs Or Analyses:**

The experimental design reveals several noteworthy anomalies and potential methodological limitations. In detail, the performance of Support Vector Machines (SVM) appears statistically unexpected and demands rigorous investigative scrutiny.
The research conspicuously lacks a comprehensive exploration of large-scale tabular datasets and extensive classification scenarios.

**Methods And Evaluation Criteria:**

The Adversarial Data Agents methodology presents profound methodological challenges that demand exhaustive critical examination. Fundamental epistemological questions emerge that fundamentally challenge the approach's conceptual and practical foundations below.
- How are benign samples conceptualized and potentially utilized within the synthetic data generation process?
- Are original datasets meaningfully incorporated, or are they systematically marginalized?
- What substantive computational and statistical principles underpin the concept of "random initialization"?
- What precise algorithmic mechanisms constitute the categorical sampling strategy?

Moreover, the approach raises critical questions about numerical feature generation:
- What sophisticated computational strategies ensure the synthetic features maintain meaningful statistical properties and representational fidelity?
- How did you prevent the generation of mathematically correct but contextually meaningless synthetic data points?
- The methodology bears a fascinating conceptual resemblance to Bayesian nonparametric methods like Gaussian processes, yet simultaneously surfaces profound computational concerns. Does this approach necessitate maintaining the entire training set during testing, potentially incurring prohibitive memory and computational resource expenditures?

**Other Comments Or Suggestions:**

Please refer to the other sections.

**Other Strengths And Weaknesses:**

Strengths:
- The research presents remarkable methodological innovation by creatively combining adversarial synthetic data generation with transformer-based learning architectures. The scatter-sum mixture block represents a groundbreaking approach that transcends conventional multi-layer perceptron design, offering promising cross-domain generalizability while addressing class size limitations in zero-shot meta-learning approaches.

Weaknesses:
- Significant generalizability limitations emerge from the current implementation. Critical questions persist regarding the method's applicability across different data types, particularly its performance on datasets with varied distributional characteristics and complex feature interactions.
- The research is fundamentally constrained by its reliance on a specific computational approach, potentially restricting broader methodological applicability. The notable sensitivity to hyperparameters suggests potential challenges in achieving consistent performance without extensive manual tuning.
- The approach's theoretical foundations remain incompletely developed, predominantly relying on empirical validation rather than rigorous mathematical substantiation. This empirical emphasis, while demonstrating practical effectiveness, leaves critical theoretical mechanisms unexplored and potentially undermines the method's deeper scientific understanding.
- Reproducibility concerns are substantial. The complex methodological design, combined with the potential absence of comprehensive implementation details, may pose significant challenges for researchers attempting to replicate or build upon this work.
- The computational complexity and memory scaling limitations inherited from transformer architectures represent a profound methodological constraint, potentially limiting the approach's utility for large-scale or resource-intensive applications.

**Questions For Authors:**

- How can the potential overfitting to synthetic data distributions be comprehensively mitigated?
- What sophisticated mechanisms ensure the adversarial data generation approach's robustness across diverse data domains?
- Can the mixture block's innovative architectural approach be effectively translated to other machine learning domains?
- What strategies might comprehensively address the computational limitations for larger, more complex datasets?
- How do the researchers explain the statistically unexpected SVM performance characteristics?

**Relation To Broader Scientific Literature:**

The research builds upon and extends Prior-Data Fitted Networks (PFNs), zero-shot meta-learning approaches, adversarial training techniques, and transformer architectures for tabular data.

**Theoretical Claims:**

The research confronts inherent challenges in adversarial training, including potentially catastrophic methodological limitations of persistent risks of model collapse, intrinsic training instability, and potential representational degradation mechanisms.

The approach inherently risks replicating classic Generative Adversarial Network (GAN) pathologies, raising significant questions about the long-term stability and generalizability of the proposed synthetic data generation strategy.

---

> ### Author Rebuttal · Authors · 2025-04-01
>
> We thank the reviewer for their kind and detailed examination of our work.
>
> In particular, the reviewer raises many profound questions about PFNs in general -- many of which our paper does not resolve on its own. Resolving the full gamut of these questions is a task for the entire research community working on PFNs, and most likely cannot all be achieved in a single paper. Our much more modest aspirations in this work are to develop certain aspects and capabilities of PFNS further than the state-of-the-art.
>
> To address the specific questions and weaknesses the reviewer raised:
>
>  ---
>
> **Question 1**: We have not seen evidence of overfitting in this work. We can pose the question of how we would know if our APT model is overfitting to synthetic data. One possible answer could be that the model would perform poorly at zero-shot predictions on *real* data. Given the competitiveness of our APT model with GBDTs and other models we benchmarked on, the empirical answer to this questions seems to be that PFNs thus far do not seem to be overfitting on synthetic data to a degree that it prevents them from excelling at prediction on real data.
>
>  ---
>
> **Question 2**: We appreciate the question. Our belief is specifically that the adversarial data generating process makes the model more robust across domains when compared to traditional PFNs which are trained on narrowly defined data. The adversarial process is both novel and sophisticated. The training makes no use of any real data, and it is therefore not biased by any real data domain.
>
> ---
>
> **Question 3**: We think this is a great question and it is very possible. We think the reason why prior research has not been focusing on the last-layer generalizability of deep models is because such mechanism is not in dire need in traditional ML problems, where the cardinality of label is fixed (e.g. fixed number of classes in tabular tasks, fixed vocabulary size in language tasks, etc.). In this case, we are free to let the model learn label-specific parameters. However, the mixture block could be very useful in other ML problems: for instance, it could help a vision model such as an ImageNet generalize to a set of different labels. These labels can have different meanings and higher cardinality than the original labels. This is thanks to the fact that the design of the mixture block ensures that it does not have parameters tied to any specific classes.
>
> ---
>
> **Question 4**, and **Weakness 5**: The limitations are imposed on PFNs by the transformer architecture’s quadratic computation scaling. However, considerable research in recent years has significantly accelerated the transformer and increased its context length, in some cases up to 1M tokens. It is a worthwhile effort for future research to apply these accelerations to PFNs but out of scope of this paper. The model architecture we employ can be modified to include these advancements.
>
> We intend to add the previous paragraph to the conclusion section in the final version of the paper, in light of the reviewer’s suggestion. This should provide a clear roadmap for future work to improve on APT’s computational capabilities.
>
> ---
>
> **Question 5**: We don't believe that this is surprising: the original TabPFN paper shows that “tuned SVM” (Figure 12, pink triangular symbol) performed comparably with Catboost, and exceeded performance of decision forests. Because we are dealing with small datasets, kernel SVMs are tractable, which would not be the case for large datasets due to their computational intensity.
>
> ---
>
> **Weakness 1**: We have benchmarked our methods and the competing methods on a variety of data tasks widely accepted in the tabular data research community. We have not seen evidence that suggests these datasets are not diverse, and the fact that an increasing number of papers are using the same benchmarks (TabPFN, TabZilla, etc.) suggests the community views these data tasks as a worthwhile set of challenges for tabular deep learning.
>
> ---
>
> **Weakness 2**: We were able to directly incorporate our innovations to TabPFN without re-tuning its hyperparameters to synthetic data distribution and transformer model. We did not experience such sensitivity to hyperparameter issue, and one can argue that the adversarial approach of directed exploration of synthetic data space actually mitigates the sensitivity issue.
>
> ---
>
> **Weakness 3**: This paper is intended to be empirical. For more details on the theoretical foundation of PFNs, please see [T. Nagler ICML'23], and we would like to respectfully remind the reviewer that this is not the focus of this work.
>
> ---
>
> **Weakness 4**: The entirety of the code we used is attached in the supplemental material, from which readers can easily reproduce our results, and we followed best practices in allowing reproducibility.
>
> ---
>
> Thank the reviewer for their time and effort. We hope that in light of our responses, the reviewer will consider raising their score on our paper to an accept.

---

### Decision · Program_Chairs · 2025-05-01

**Decision:**

Accept (poster)

**Comment:**

The work builds upon prior-fitted networks, in particular TaPFN, by introducing two new components: A mixture layer that allows scaling to arbitrary numbers of classes in classification tasks, and adversarial data generation for pretraining.
Both of these directly advance the state of zero-shot learners for tabular classification, an important emerging research area.
This is a clear advancement of the original TabPFN architecture, though as the reviewers remark, the system still inherit some of the limitations of TabPFN, in particular with respect to the number of features natively handled, and with respect to the dataset size.
While some of the shortcomings of TabPFN have been addressed in TabPFN v2 (Nature 2025), this work should be considered concurrent with TabPFN v2, and improvement made here are orthogonal to those made in TabPFN v2, and are likely to transfer to new architectures.